# The Role of Cellular Stress in Intrauterine Growth Restriction and Postnatal Dysmetabolism

**DOI:** 10.3390/ijms22136986

**Published:** 2021-06-29

**Authors:** Shelby L. Oke, Daniel B. Hardy

**Affiliations:** 1Department of Physiology and Pharmacology, Schulich School of Medicine and Dentistry, Western University, 1151 Richmond Street, London, ON N6A 5C1, Canada; soke2@uwo.ca; 2The Children’s Health Research Institute, The Lawson Health Research Institute, London, ON N6A 5C1, Canada; 3Department of Obstetrics and Gynaecology, Schulich School of Medicine and Dentistry, The University of Western Ontario, 1151 Richmond Street, London, ON N6A 5C1, Canada

**Keywords:** intrauterine growth restriction (IUGR), metabolism, cell stress, cell death, metabolic syndrome

## Abstract

Disruption of the in utero environment can have dire consequences on fetal growth and development. Intrauterine growth restriction (IUGR) is a pathological condition by which the fetus deviates from its expected growth trajectory, resulting in low birth weight and impaired organ function. The developmental origins of health and disease (DOHaD) postulates that IUGR has lifelong consequences on offspring well-being, as human studies have established an inverse relationship between birth weight and long-term metabolic health. While these trends are apparent in epidemiological data, animal studies have been essential in defining the molecular mechanisms that contribute to this relationship. One such mechanism is cellular stress, a prominent underlying cause of the metabolic syndrome. As such, this review considers the role of oxidative stress, mitochondrial dysfunction, endoplasmic reticulum (ER) stress, and inflammation in the pathogenesis of metabolic disease in IUGR offspring. In addition, we summarize how uncontrolled cellular stress can lead to programmed cell death within the metabolic organs of IUGR offspring.

## 1. Introduction

The metabolic syndrome refers to a group of physiological symptoms that increase an individual’s risk for cardiovascular disease and type II diabetes. These symptoms, including dyslipidemia, obesity, hyperglycemia, and hypertension, are often assessed independent of each other; however, their simultaneous occurrence is synergistic toward onset of the metabolic syndrome. It is well known that these symptoms are influenced by factors such as genetics and lifestyle, but the role of developmental priming is often overlooked. The developmental origins of health and disease (DOHaD) posits that there is an inverse relationship between birth weight and long-term metabolic health, as adverse events in utero may permanently influence the function of metabolic organs. Infants affected by intrauterine growth restriction (IUGR) exhibit impaired organ growth with metabolic disease in adulthood, as early epidemiological studies by Sir David Barker and colleagues determined that low birth weight individuals have high rates of obesity, glucose intolerance, and coronary artery disease [1,2,3]. These studies have since led to widespread investigation of the underlying causes of IUGR, as well as the metabolic pathologies that arise in response to impaired organ development.

IUGR occurs as a consequence of utero-placental insufficiency, whereby the placenta does not meet metabolic requirements set by the fetal genome [4]. When placental tissue is unable to support proper nutrient and oxygen exchange, select fetal organs exhibit reduced growth and decreased cell size [5]. This ‘organ-sparing’ effect occurs such that vital organs (i.e., the heart and brain) receive greater shares of available resources at the expense of other organs, such as the liver [6]. Utero-placental insufficiency is often secondary to insults of maternal origin, including maternal malnutrition, drug use, and infection among others; therefore, maternal lifestyle has a significant influence on offspring health [4,5]. Importantly, the postnatal environment has also been demonstrated to play an indirect role in provoking long-term metabolic dysfunction, as offspring born into an environment that is ‘mismatched’ from that in utero are subject to maladaptive changes in fetal programming [7]. Animal studies have revealed several mechanisms that govern this relationship, including epigenetic regulation of gene expression, the microbiome, and the hypothalamus-pituitary-adrenal axis. In addition, cellular stress is known to have a major role in causing adverse metabolic health across various models of IUGR. 

Fetal growth and development consist of intricate cellular processes that are highly sensitive to intra- and extracellular stressors. Because of this, it is plausible that the presence of metabolic disease in adult IUGR offspring is attributed in part to cellular stress and programmed cell death. While events of cell stress and cell death are often protective, they can also be destructive and contribute to the development of metabolic disease. Studies have demonstrated that a suboptimal prenatal environment initiates cell stress and cell death in the placenta, giving rise to compromised fetal growth. This may further lead to cellular stress and dysfunction in metabolic organs, including oxidative stress and mitochondrial dysfunction, endoplasmic reticulum (ER) stress, inflammation, apoptosis, and autophagy. Alternatively, the occurrence of rapid postnatal weight gain (i.e., catch-up growth) in low birth weight offspring can lead to cellular stress and metabolic disease in an indirect manner. As such, this review addresses the latest research findings on the various types of cellular stress and programmed cell death that are prevalent in IUGR offspring during postnatal life. 

## 2. Human Studies of Intrauterine Growth Restriction and Metabolic Outcomes

Evidence for DOHaD has been revealed by numerous human studies, many of which are focused on the role of famine and other nutritional insults during pregnancy. Barker’s investigations of the Hertfordshire birth cohort are perhaps the best described, as these studies were among the first to uncover the relationship between impeded fetal growth and adult offspring health [8]. In combination with epidemiological studies of famine, the data from this cohort indicate a strong inverse relationship between birth weight and risk for the metabolic syndrome. In particular, men and women who were born with low birth weight had increased risk of death from cardiovascular disease in adulthood, while this was also true of men who were small at one year of age [8]. These studies led to further investigation of the surviving individuals from the cohort, and it was found that low birth weight was associated with increased risk for insulin resistance [9] and type II diabetes [10], as well as coronary artery disease [11]. These men and women also had increased risk for osteoporosis [12] and sarcopenia [13], indicating that the effects of small size at birth were not limited to just metabolic organs. Additional birth cohorts from Brazil, South Africa, and the United States have also reported similar trends, further solidifying the role of the in utero environment in the development of metabolic disease [14,15,16]. 

As mentioned previously, Barker’s studies of the Dutch famine cohort have brought much attention to the importance of a high-quality maternal diet in determining offspring metabolic health. The data, which were collected from individuals born before, during and after the famine of 1944–1945, demonstrated that those exposed to famine during mid to late gestation were glucose intolerant as adults [2]. Later studies of this cohort found that exposure to famine at any point during pregnancy also contributed to increased likelihood of hypertension during adult life [17]. Studies of the Chinese famine of 1959–1961 have reported similar trends, as fetal exposure to famine resulted in elevated fasting plasma glucose in adulthood [18]. Additionally, a retrospective cohort study of Chinese individuals found that risk for hypertension was increased in offspring exposed to famine exclusively during the first trimester [19]. Many studies have determined that exposure during infancy or childhood appears to play a much greater role in this process [18,20], highlighting the importance of the postnatal environment on offspring metabolic health and the complex relationship between fetal development and metabolic disease. 

Developmental programming also provides an alternative explanation for rising rates of obesity, as there is an association between birth weight and adult body composition. This relationship is U-shaped such that infants born with either IUGR (low birth weight, <2500 g) or macrosomia (high birth weight, >3500 g) have increased risk for obesity in adult life compared to those born appropriate for gestational age [21]. Data from the Dutch famine cohort indicate that males exposed to famine during the first two trimesters were more likely to be obese during adulthood [1], while female offspring exposed to the Chinese famine experienced a similar risk [22]. Alarmingly, obesity and dyslipidemia are risk factors for insulin resistance and type II diabetes (as reviewed by James et al. (2004) [23]; therefore, these individuals are at further risk for the metabolic syndrome. The previously mentioned birth cohorts from Brazil, South Africa, and the United States have also outlined the role of postnatal catch-up growth in elevating the risk for obesity. In these studies, increased weight gain during childhood was associated with increased fat mass and weight gain during mid-childhood and adolescence [14,15,16]. Importantly, childhood obesity is considered to be further predictive of adult body mass index and body composition [24]; therefore, the battle against obesity and the metabolic syndrome is lifelong.

In addition to excess visceral adiposity, impaired liver function is critical in contributing to dyslipidemia. The liver is central in managing systemic levels of nutrients in both the fed and fasted state, as it coordinates the various metabolic pathways that maintain appropriate levels of glucose and lipid molecules. Individuals with hepatic dysfunction exhibit hyperlipidemia (i.e., elevated triglycerides and cholesterol), which can give rise to fibrosis and eventual cirrhosis of the liver [25]. Importantly, this may further lead to glucose intolerance and the development of type II diabetes, indicating the high potential for hepatic dysfunction in producing widespread metabolic disease [25]. Barker’s early studies identified that IUGR offspring exhibit asymmetrical growth patterns throughout pregnancy; these offspring have impaired liver growth as indicated by decreased abdominal circumference at birth [6]. These same offspring display elevated levels of total and low-density lipoprotein (LDL) in adult life, and the strong correlation between these two variables suggest that long-term hepatic function becomes compromised following gestational insult. Again, IUGR infants undergo rapid catch-up growth in early postnatal life, and the undernourished liver stands to gain the most relative to other organs [26,27]. This is evident in small for gestational age infants who undergo hypersomatotropism by postnatal day (PND) four, which occurs due to increased levels of insulin growth factor 1 (IGF-1) in the liver and blood [28]. Based on Barker’s ‘thrifty phenotype’ hypothesis, which states that fetal programming will adapt in anticipation of a similar postnatal environment, this rapid weight gain is detrimental to hepatic function and can further exacerbate risk for the metabolic syndrome [7]. In fact, preterm infants undergo catch-up growth following consumption of nutrient-dense formula, leading to high ratios of LDL to high density lipoprotein (HDL) [29]. Finally, cohort studies of Finnish and Italian infants have identified an association between birth weight and risk for non-alcoholic fatty liver disease (NAFLD) during childhood and adult life [30,31]. Again, this increased risk for NAFLD may contribute to later onset of insulin resistance and cardiovascular disease [25], warranting the need for clinical preventative strategies in improving fetal growth and development. 

While the clinical evidence in support of DOHaD is abundant, these studies are limited in that they provide little insight into the underlying mechanisms of these physiological observations. It is for this reason that animal models are essential in improving our understanding of how cellular stress and programmed cell death contribute to adverse metabolic outcomes of IUGR offspring. Given that there are a variety of causes of IUGR, many different animal models have been established in order to investigate DOHaD. These include, but are not limited to, nutritional models of IUGR (e.g., maternal caloric restriction, maternal protein restriction, maternal obesity), maternal drug exposure (i.e., nicotine, anti-depressants, and/or cannabinoids), maternal infection, maternal hypoxia, and maternal stress (i.e., exposure to elevated glucocorticoids). Additionally, surgical intervention can be used to generate models of uterine ligation, which mimic the effects of placental insufficiency by reducing both nutrient and oxygen transfer to the fetus. These models are advantageous in that they allow for assessment of not only the direct effects of in utero insults on cellular stress and death, but the effects of indirect insults as well (i.e., postnatal catch-up growth). In this review, we outline the research performed to date investigating the role of cellular stress and death in some of these aforementioned models. 

## 3. Cellular Stress and Metabolism in IUGR Offspring

### 3.1. Oxidative Stress and Mitochondrial Dysfunction

Pregnancy and early postnatal development are metabolically challenging periods of growth for both the mother and offspring. Accordingly, IUGR often causes increased stress within mitochondria, as this organelle has a critical role in energy production. Mitochondria utilize oxygen as the terminal electron acceptor in aerobic respiration; however, the production of damaging reactive oxygen species (ROS) and ROS by-products is a negative side effect [32]. While low levels of ROS are required for some enzymatic reactions and signaling pathways, they inflict oxidative damage on macromolecules when present in excess [33]. Oxidative stress arises when there is an imbalance between ROS production and antioxidant enzymes, which are responsible for the transformation of ROS into less harmful molecules [32]. The mitochondrial electron transport chain is both a source and target of ROS; therefore, mitochondrial dysfunction commonly coexists with unabated oxidative stress [32]. Importantly, both oxidative stress and mitochondrial dysfunction are consistently associated with the IUGR-induced metabolic syndrome. Studies have revealed that mothers of growth-restricted offspring tend to have increased ROS and decreased levels of antioxidants in the blood [34,35,36,37], while tissues taken from IUGR offspring also demonstrate oxidative stress. This includes not only an increase in ROS, but also the differential expression of antioxidant enzymes, increased lipid peroxidation, and compromised synthesis of ATP (as reviewed by Rodriguez-Rodriguez et al., 2018) [38]. Collectively, these trends indicate that oxidative stress is an important contributor to the aberrant metabolism seen in IUGR offspring. What follows is a detailed discussion of how oxidative stress and mitochondrial dysfunction in the growth-restricted liver, pancreas, adipose, skeletal muscle, heart, and kidney play a role in onset of the metabolic syndrome.

The liver is highly subjected to oxidative stress given that it is abundant in mitochondria, due to its critical role in nutrient metabolism. Many chronic forms of liver disease, including non-alcoholic steatohepatitis (NASH) and NAFLD, are characterized by increased markers of oxidative stress, which is often accompanied by an accumulation of damaged or dysfunctional mitochondria [39,40,41,42]. The IUGR liver is no exception to this, as numerous studies have found evidence of hepatic oxidative and mitochondrial stress in growth-restricted offspring. For example, in a porcine model of spontaneous IUGR, growth-restricted neonates had increased levels of hepatic alpha-1-acid glycoprotein at birth, indicating hepatic and systemic oxidative stress [43]. These offspring also displayed increased protein levels of complex IV of the electron transport chain, suggesting that ATP availability may be reduced due to accelerated ATP hydrolysis [43]. In a rat model of caloric restriction, a 50% decrease in maternal caloric intake from gestational day 11 through PND 21 (i.e., the point of weaning) resulted in increased lipid peroxidation marker 4-hydroxynonenol (4HNE) and decreased glutathione protein levels at three weeks [44]. Adult offspring had unaltered 4HNE and antioxidant levels after switching to a control diet at weaning, suggesting that the caloric restriction itself had caused the oxidative stress [44]. Many studies using the maternal protein restriction (MPR) rat model have also established that catch-up growth may be of greater detriment to hepatic mitochondrial function rather than the original in utero nutritional insult. In the MPR model, pregnant rat dams are fed a low protein diet throughout gestation, thereby inducing fetal protein deficiency and IUGR. As seen in Figure 1A, offspring that are maintained on a low protein diet postnatally (LP1 offspring) do not undergo catch-up growth and remain small relative to control offspring. These offspring are metabolically healthy in adult life, while they also exhibit decreased hepatic ROS production [45,46]. Given that LP1 offspring do not experience a nutritional mismatch between the pre- and postnatal environments, it is possible that the maintenance of the low protein diet may be protective of mitochondrial function in the absence of catch-up growth. Alternatively, whole body and hepatic catch-up growth is induced by switching offspring to a normal protein diet at weaning (LP2 offspring) or at birth (LP3 offspring) [45]. We have shown that LP2 offspring exclusively develop hypercholesterolemia, glucose intolerance, and impaired drug metabolism at four months of age, while LP1 and LP3 offspring appear to be unaffected [45,47]. At the same time, LP2 offspring exhibit aberrant markers of aerobic metabolism (e.g., increased lactate dehydrogenase and phosphorylated pyruvate dehydrogenase, decreased citrate synthase and complex II), increased superoxide dismutase 1 (SOD1) and SOD2, decreased catalase, and increased lipid peroxidation (i.e., 4HNE) in the liver (Figure 1B and Figure 2) [48]. LP2 offspring also displayed increased protein abundance of p66Shc (Figure 1B), an adaptor protein that can accelerate the production of ROS through binding with cytochrome C [48]. This was demonstrated to occur as a direct result of ER stress, which is known to exist simultaneously with oxidative stress and mitochondrial dysfunction [49]. Other studies have also shown that catch-up growth after weaning results in increased hepatic oxygen consumption, altered antioxidant level and activity, and decreased mitochondrial DNA-encoded gene expression [46,48,50,51]. Again, LP3 offspring do not exhibit these metabolic and mitochondrial deficits, so the timing of nutritional restoration and catch-up growth appear to be of importance in initiating oxidative stress as a driver of poor liver health. By introducing a normal protein diet prior to the completion of hepatic differentiation, the liver may be protected from the metabolic consequences of oxidative stress and poor mitochondrial function.

The IUGR pancreas, adipose, and skeletal muscle are also subject to oxidative stress and mitochondrial dysfunction in postnatal life. The presence of oxidative stress in these organs is thought to be a major contributor to the development of insulin resistance and diabetes, as ROS-mediated damage can interfere with glucose storage and metabolism. The human IUGR pancreas presents with β-cell dysfunction and impaired insulin secretion at birth [52], and this can persist throughout childhood and adulthood [53,54,55]. Animal studies have implicated oxidative stress in this process, whereby IUGR pancreatic islets experience a gradual increase in the production of ROS over time [56]. Islets isolated from rat IUGR offspring at one week of age demonstrated an increase in dichlorofluororescin fluorescence (i.e., ROS), and this was increased further in islets isolated at 15 weeks such that fluorescence was more than two-fold higher when compared to control islets [56]. In addition, the activity of citrate synthase and complexes I and III were decreased at 15 weeks of age, along with decreased production of ATP throughout fetal and postnatal life [56]. Studies of male protein-restricted offspring also show that ROS production is increased in pancreatic islets at three months of age, as well as increased transcript abundance of complexes I and III [57]. Not only does this provide evidence for pancreatic mitochondrial dysfunction, but also that mitochondrial respiration may be compromised in a sex-specific manner such that females are protected from oxidative damage [57]. The protein-restricted pancreas also exhibits altered expression and activity of SOD, catalase, and glutathione peroxidase throughout postnatal life [50]; therefore, discrepancies in antioxidant capacity may serve as an underlying cause of oxidative stress in the postnatal IUGR pancreas. Similarly, glucose uptake and metabolism are greatly affected by oxidative stress and mitochondrial dysfunction in IUGR skeletal muscle. Rodent models of uteroplacental insufficiency have found that insulin-stimulated glucose uptake and glycogen content are reduced in skeletal muscle of IUGR offspring during early adulthood, consistent with decreased ATP production and activity of enzymes involved in aerobic metabolism [58,59]. Similar results were found in a porcine model of IUGR, whereby growth-restricted piglets had increased susceptibility to mitochondrial dysfunction in skeletal muscle following exposure to a high-fat diet in postnatal life [60]. In a rodent model of maternal caloric restriction, undernourished offspring also displayed decreased mitochondrial number and content at ten weeks of age [61]. This was consistent with decreased coupled and uncoupled respiration (states 3 and 4, respectively), decreased fatty acid oxidative capacity, and decreased respiration through complex I [61]. Interestingly, protein-restricted rat offspring show increased antioxidant abundance in skeletal muscle, concomitant with altered activity of aerobic metabolism enzymes [62]. Gestational exposure to nicotine has been found to elicit similar effects in adipose tissue at three weeks and six months of age, as SOD1 and SOD2 protein levels were 37–48% higher in the white adipose tissue of IUGR offspring compared to control offspring [63]. While the increase in antioxidant abundance is thought to occur as a compensatory mechanism in response to oxidative stress, it has also been proposed that inflammation may have a contributing role in this process [62,63].

The long-term function of cardiovascular organs is also impaired by oxidative stress due to fetal growth restriction. The mammalian heart possesses limited regenerative capacity in postnatal life, and the presence of mitochondrial-dependent oxidative stress has been demonstrated to shorten the postnatal proliferative period in mouse offspring [64]. For example, exposure to maternal nicotine in utero results in decreased protein disulfide isomerase (PDI), which is important in protecting the heart from ischemic insult [65]. Deficiencies in PDI can result in oxidative and mitochondrial damage, so it is not surprising that maternal nicotine exposure also causes decreased protein abundance of cardiac SOD2 and mitochondrial complexes I, II, IV and V [65]. Of note, these changes occurred exclusively in three-month-old offspring after catch-up growth had occurred, suggesting that catch-up growth may be responsible for the observed cardiac deficits. Nephrogenesis and renal function are also impeded in IUGR offspring via reduced nephron endowment, making these offspring further susceptible to hypertension and chronic kidney disease [66]. Studies of maternal malnutrition have identified the protein fetuin-B as having a role in this process, leading to increased generation of ROS in the kidneys of low birth weight offspring [67]. Much like the IUGR pancreas, the programming of mitochondrial dysfunction and oxidative stress also appears to be sex-dependent in the kidney. Woodman et al. showed that prenatal iron deficiency leads to decreased mitochondrial content and respiration exclusively in the male kidney, along with increased SOD and nitric oxide levels [68]. Male iron-deficient offspring also exhibit reduced systolic blood pressure in adult life, and it is believed that this occurs in response to impaired metabolic function in the male kidney [69].

### 3.2. Endoplasmic Reticulum (ER) Stress and the Unfolded Protein Response

As mentioned previously, oxidative stress is also known to occur alongside endoplasmic reticulum (ER) stress. The mitochondrion and ER are physically connected at sites called the mitochondrial-associated ER membrane (MAM); these sites can indirectly influence the production of ATP, and they are responsive to ER signaling during instances of increased protein folding [70]. When the ER cannot facilitate proper protein folding, this leads to lumenal accumulation of misfolded or unfolded proteins in the ER (i.e., ER stress). While ER stress is essential for embryonic development and the maintenance of pregnancy [71], excessive or chronic ER stress during postnatal life can be triggered by numerous cellular insults. These may include oxidative stress, diminished calcium homeostasis, decreased supply of amino acids, viral infection, decreased formation of disulfide bonds, and decreased N-linked glycosylation [72,73]. Furthermore, postnatal catch-up growth has been shown to trigger ER stress, specifically in the growth-restricted liver [74]. In the unstressed cell, the chaperone protein Grp78 prevents the activation of ER transmembrane proteins IRE1, PERK, and Atf6 by binding their N-terminal domains in the ER lumen [75,76]. Under ER stress, the cell’s unfolded protein response (UPR) becomes activated via release of these transmembrane proteins from Grp78 [75,76]. This allows for their oligomerization and subsequent activation of targets that promote an adaptive phenotype [75]. This entails increased transcription of genes that strengthen the folding capacity of the ER, or protein degradation. If the ER cannot be successfully relieved of misfolded and unfolded proteins, ER-stress induced cell death is initiated through the activation of C/EBP-homologous protein (CHOP) [72,73]. Certain ER stress pathways have been implicated in the onset of various metabolic diseases (as reviewed by Cnop et al., 2012) [77], so the presence of ER stress in IUGR tissues is not surprising. 

Various animal models have found that hepatic ER stress may serve as an underlying mechanism for the dysregulated blood glucose and insulin resistance exhibited by IUGR offspring. As mentioned previously, the liver regulates metabolic pathways that maintain blood glucose levels, including gluconeogenesis. Rodent models demonstrate that activation of the UPR either precedes or occurs simultaneously with accelerated hepatic gluconeogenesis, and these offspring later exhibit glucose intolerance and impaired hepatic glycogen storage [47,74,78,79]. For example, one study showed that uteroplacental insufficiency due to uterine artery ligation resulted in increased markers of ER stress in the affected pups at birth, concomitant with increased transcript abundance of gluconeogenic enzymes Phosphoenolpyruvate Carboxykinase 1 (PCK1) and glucose-6-phosphatase catalytic subunit (G6Pc) [79]. At PND 105, these male IUGR offspring were glucose intolerant and still had active markers of ER stress [79]. The UPR is also activated in protein-restricted offspring via upregulation of Atf2, Atf6, and phosphorylated eukaryotic transcription factor 2 alpha (p-eIF2α) during fetal life, followed by elevated fasting blood glucose and decreased Periodic acid-Schiff (PAS) stain (i.e., decreased glycogen storage) at twelve weeks [78]. Alternatively, Sohi et al., demonstrated that ER stress occurs exclusively following catch-up growth in the protein-restricted liver (i.e., LP2 offspring) [74]. These LP2 offspring exhibited increased hepatic Grp78, Grp94, p-eIF2α[Ser51], and spliced X box binding protein 1 (Xbp-1; Figure 1B), along with increased protein levels of p85 and decreased phosphorylation of protein kinase B (Akt1) at serine residue 473 (p-Akt1[Ser473]) [74]. Increased p85 and decreased p-Akt1[Ser473] are commonly associated with impaired insulin signaling, while p-Akt [Ser473] and p-eIF2α[Ser51] are also inversely related and occur with insulin resistance [74,80]. Again, LP2 offspring exhibit increased p66Shc protein levels indicating oxidative stress (Figure 1B), and in vitro studies suggest that this may occur as a direct result of ER stress [48]. Of note, offspring that had not undergone catch-up growth (i.e., LP1 offspring) did not exhibit indices of ER stress, oxidative stress, or insulin resistance [48,74]. Protein levels of Grp78 and Grp94 were also increased at embryonic day (ED) 19; therefore, it is possible that hepatic ER stress begins in the protein-restricted liver during fetal life and is worsened with postnatal catch-up growth [74]. 

Given that ER stress is linked to glucose intolerance and insulin resistance, it seems appropriate that IUGR offspring exhibit activation of the UPR in additional organs involved in glucose homeostasis. Much like the liver, gestational exposure to a low protein diet leads to activation of all three UPR pathways in the pancreas; however, this was exclusive to the fetal pancreas at ED20 [81]. Fetal pancreata displayed increased Atf4/6, protein kinase R-like endoplasmic reticulum kinase (PERK), Xbp-1, p-eIF2α, and cAMP responsive element-binding protein-3-like 3 (Creb3l3), while adult offspring only had upregulation of Atf6 and Creb3l3 [81]. An additional study found that Grp78 was increased in fetal pancreatic islets following protein restriction, along with increased transcript and protein levels of CHOP [82]. This was associated with increased levels of other pro-apoptotic proteins in the fetal IUGR pancreas, suggesting that ER stress-induced cell death may be present [82]. This makes sense considering that the protein-restricted fetus also exhibits deficient levels of insulin promoter factor 1 (Pdx1), which increases the susceptibility of β-cells to ER stress-induced apoptosis [83]. Collectively, it is possible that the prolongation of ER stress and resulting β-cell apoptosis are responsible for the insulin resistance of protein-restricted offspring. Finally, Fritz et al. have implicated endoplasmic reticulum-localized DnaJ 4 (ERdj4), a Grp78 co-chaperone, in causing pancreatic ER stress and IUGR [84]. Genetic deletion of ERdj4 in mice resulted in low birth weight offspring with reduced glycogen stores and hypoglycemia at birth, while surviving adult offspring displayed constitutive ER stress in the pancreas [84]. Adult offspring also experienced β-cell loss, glucose intolerance and impaired insulin synthesis [84]. 

ER stress is further present in the adipose tissue of IUGR offspring, as seen in rats born from a model of uteroplacental insufficiency [85]. At PND21, male growth-restricted offspring had increased transcript levels of Grp78, Atf6, p-eIF2α, and CHOP, followed by glucose intolerance at PND45 [85]. Conversely, female IUGR offspring born from uteroplacental insufficiency have elevated ER stress markers in white adipose tissue at an advanced age [86]. While it is possible that ER stress occurs in a sex-specific manner, it is unknown whether these differences are consistent across the entire lifespan of rodents as these two studies assessed markers of ER stress at vastly different time points. 

### 3.3. Inflammation and the Immune Response

The immune system is the body’s first line of defence against pathogens, damaged cells, and toxic environmental contaminants. It is composed of multiple cell types that provide protection against infection, and these cells elicit an inflammatory response by releasing pro-inflammatory cytokines, chemokines, and growth factors. Although a necessary biological response, acute inflammation can lead to tissue damage and disease when left unresolved. Inflammatory pathways become upregulated in patients with the metabolic syndrome; therefore, chronic inflammation is considered to be a key contributor in the pathogenesis of metabolic disease. In particular, inflammation has been demonstrated to interfere with endothelial cell function as an underlying cause of cardiovascular disease [87]. While this process remains poorly understood, the immune system has become an attractive therapeutic target in the treatment of the metabolic syndrome. 

There has been some interest in the role of inflammation in DOHaD, as an impaired immune response may predispose growth-restricted individuals to metabolic pathologies. Rodent IUGR offspring exposed to various doses of maternal nicotine have elevated serum levels of pro-inflammatory cytokines at birth, including interleukin (IL)-6, transforming growth factor β (TGFβ), and tumor necrosis factor alpha (TNFα) [88]. The authors further reported an increase in serum C-reactive protein and nitrite oxide, which are also considered to be markers of inflammation [88]. Studies of preterm piglets have found that growth-restricted offspring exhibit increased serum IL-10 at birth; however, this increase was no longer present by PND19 [89]. That said, total leukocyte and neutrophil counts were elevated in the IUGR group at PND 8–10 [89]. Obesity is commonly associated with inflammation; therefore, it is not surprising that pro-inflammatory markers are also upregulated in IUGR offspring with increased adipose deposition. Uteroplacental insufficiency results in increased levels of TNFα in the serum and subcutaneous adipose tissue of male offspring at three weeks of age, followed by glucose intolerance at PND45 [85]. Finally, immunohistochemical analyses have found heightened expression of TNFα in the livers of IUGR piglets at PND7, together with an increase in the ratio of Kupffer cells to hepatocytes [90]. Moreover, these offspring had significant differences in the hepatic proteomic profile of enzymes involved in carbohydrate metabolism, protein metabolism, and oxidative stress [90]. Studies of maternal protein restriction further demonstrate that rodent IUGR offspring have augmented circulating and hepatic cholesterol in adult life, which may be in part due to increased TNFα in the fetal and adult liver [45,91]. Interestingly, a study of adult IUGR pigs found that dietary supplementation of resveratrol from PND7–21 improved inflammation and mitochondrial dysfunction in the liver, thereby alleviating hepatic lipid accumulation [92]. That said, additional studies are needed to clarify the relationship between inflammation and mitochondrial function in causing metabolic disease in IUGR offspring. 

## 4. Programmed Cell Death and Metabolism in IUGR Offspring

As mentioned previously, cell stress often occurs in a protective manner. That said, an organism’s response to cell stress is dependent on the type and severity of the gestational insult. When an insult is severe and/or chronic enough that the resulting cell stress is overwhelming, cellular death may occur. Programmed cell death is a controlled cellular response that works to eliminate damaged or dysfunctional cells, either by means of apoptosis or autophagy. 

### 4.1. Apoptosis

Apoptosis is a highly regulated form of cell death consisting of morphological changes that are distinct from other forms of cell death. Apoptotic cells undergo cell shrinkage, membrane blebbing, and fragmentation of nuclei and chromatin, often without inducing inflammation. This is what distinguishes apoptosis from necrosis, a type of uncontrolled cell death that provokes an immune response. While apoptosis is an essential part of many developmental processes, it is also a large contributor to the impaired function of metabolic organs. For example, the apoptotic loss of pancreatic β-cells is believed to be a driving factor of type II diabetes, particularly after the pancreas experiences oxidative stress or ER stress [93]. Although apoptosis does not cause inflammation, it has been demonstrated to occur as a result of upregulated immune cell function and pro-inflammatory cytokines (as reviewed by Quan et al., 2013) [94]. Finally, the loss of cardiomyocytes via apoptosis may also contribute to chronic heart failure, as early studies have shown that apoptosis exists in myocardial tissue samples taken from patients following myocardial infarction, cardiomyopathy, and end-stage heart failure [95,96,97,98]. Taking all of this into account, the occurrence of apoptosis in IUGR offspring seems to be cause for concern when assessing for risk of the metabolic syndrome. 

While there is considerable evidence that apoptosis in the placenta contributes to IUGR, the role of apoptosis in the fetus and postpartum is not as well understood. The IUGR heart seems to be particularly influenced by apoptosis, which is concerning due to the lack of regenerative capacity of cardiomyocytes. Hypoxia-induced growth restriction has been found to promote increased apoptosis of cardiomyocytes in E20 chick embryos, alongside poor cardiac function and cardiomyopathy in adult life [99]. Rodent models of maternal hypoxia and maternal nutrient restriction have also found decreased numbers of cardiomyocytes in adult IUGR offspring, and it is likely that this occurs as a result of increased apoptosis [100,101,102]. Adult offspring exposed to maternal hypoxia specifically display increased apoptosis and decreased cardiac power following ischemia-reperfusion injury, suggesting that the IUGR heart does not fully recover following myocardial infarction [101,102]. At the same time, the injured cardiomyocytes begin to rely on glycolysis for the production of ATP rather than the preferred method of fatty acid oxidation [102]. While it is unclear whether apoptosis has a causative role in this metabolic inflexibility, the impairments in cardiac function of IUGR offspring suggest that apoptosis is involved in conferring increased risk for chronic heart failure. 

Apoptosis is also present in the IUGR pancreas and liver, leading to abnormal development and function of these organs. Both the IUGR pancreas and liver are especially sensitive to diet-induced apoptosis, as many models of maternal undernutrition have shown elevated rates of apoptosis in pancreatic β-cells and hepatocytes. Fetal pancreatic islets simultaneously experience high rates of apoptosis and decreased proliferation following maternal protein restriction, resulting in decreased β-cell mass, reduced insulin production, and impaired insulin secretory capacity at birth [103,104,105]. Similar trends have been found in the liver following maternal caloric restriction; hepatocytes isolated from growth-restricted sheep have a reduced proliferation index, increased apoptotic cell number, and increased activity of pro-apoptotic enzymes [106]. While these studies demonstrate that apoptosis occurs during fetal development, there is also evidence for its continuation during postnatal life. Petrik et al. showed that 15% of pancreatic islet cells in protein-restricted offspring were positive for TUNEL staining at PND14, compared to 8% of islet cells in offspring fed a control diet [104]. Using a rodent model of gestational diabetes mellitus (GDM), Luo et al. also found increased protein expression of apoptotic markers in hepatocytes isolated from GDM offspring at 8 weeks of age [107]. Of note, gestational supplementation of the amino acid taurine has been shown to attenuate apoptosis in both the IUGR pancreas and liver [103,107]. Taurine is an endogenously expressed amino acid found in meat-based protein, and it is thought to mitigate apoptotic stimuli such as inflammation and mitochondrial damage [108,109,110]. While the protective role of taurine remains relatively unclear, it is possible that supplementation of taurine during pregnancy may help in preventing poor development of the pancreas and liver through reduced apoptosis [103].

### 4.2. Autophagy

Autophagy, also known as “self-eating”, is a form of programmed cell death that aims to clear out damaged organelles, misfolded or aggregated proteins, and intracellular pathogens. Following the identification of unwanted intracellular cargo, an isolation membrane called the phagophore engulfs cell material to form the autophagosome structure. Through fusion with lysosomes, the autophagosome then becomes an autolysosome and permits enzymatic degradation of its contents. While highly complex and not as well understood as apoptosis, autophagy is important in the balance of energy sources during development or times of metabolic stress. Because of this, activation of the AMP-activated protein kinase (AMPK) pathway is recognized as being a promotor of autophagy, while the mammalian target of rapamycin (mTOR) pathway is inhibitory. Dysregulated autophagy in the pancreas and liver has been previously associated with obesity and diabetes, and it is for this reason that autophagy has become of interest in the field of DOHaD. 

Much like apoptosis, most studies to date concerning the developmental consequences of autophagy tend to focus on its role in the placenta. That said, there has been some investigation of autophagy in the IUGR heart, pancreas, and liver. In the fetal baboon heart, autophagy occurs as a result of maternal caloric restriction [111]. Furthermore, male IUGR offspring display increased levels of autophagy-related 7 (ATG7) protein and LC3BII, along with fibrosis of the left ventricle [111]. It was proposed that autophagy may be upregulated in efforts to protect the heart from metabolic reprogramming and oxidative stress; however, this prediction was not investigated further in postnatal life. Moreover, both autophagy and ventricular fibrosis were absent in female offspring, indicating that autophagy may be a sexually dimorphic mechanism in the heart. In the protein-restricted rat pancreas, autophagy markers LC3BII, Beclin1, and p-ULK1 are upregulated at birth [82]. Examination of pancreata by transmission electron microscopy further revealed the presence of autophagosomes in islet β-cells, while mTORC protein levels were also decreased relative to control offspring [82]. Conversely, another study found LC3BII and Beclin1 protein abundance to be decreased in the IUGR fetal pancreas following protein restriction [81]. These findings were not expected by the authors of the study; however, they speculated that autophagy may have been suppressed due to decreased phosphorylation of AMPK [81]. It was also noted that the whole pancreas was used for molecular analyses, and the authors acknowledged that the endocrine and exocrine pancreas are affected differently by the in utero environment. 

In the IUGR liver, perinatal nutritional status seems to have a critical role in regulating autophagy. In newborn rat offspring exposed to caloric restriction during the final week of gestation, hepatic autophagy was inhibited by increased activation of mTORC1 and decreased activation of AMPK [112]. This occurred alongside hyperinsulinemia and poor glycogen mobilization immediately after birth, while the inhibition of hepatic autophagy was found to occur as a result of defective glucagon signaling [112]. Another study investigating the effects of early postnatal caloric restriction found that adult male offspring have increased hepatic autophagy following the restoration of a normal diet at PND 21, again due to activation of the AMPK pathway [44]. Prior to nutritional restoration, these offspring displayed decreased markers of autophagy and downregulation of AMPK; however, this resulted in increased polyubiquitination of proteins and hepatic oxidative stress [44]. Given that oxidative stress was alleviated in adult offspring, it could be that autophagy has a protective role in the adult IUGR liver [44]. That said, very few studies have examined the role of autophagy in postnatal life, and further studies are necessary in determining if postnatal catch-up growth leads to increased autophagy, particularly in a sex-dependent manner.

## 5. Conclusions

Epidemiological studies have provided astonishing evidence for the role of developmental programming in affecting susceptibility to the metabolic syndrome. It is clear that the perinatal period is a critical window for fetal reprogramming, while the early postnatal environment also has influence on offspring metabolic health. Animal studies have established that the inverse relationship between birth weight and long-term metabolism is mediated by mechanisms of cell stress, including oxidative stress, mitochondrial dysfunction, ER stress, and inflammation. When cell stress goes unresolved, this can lead to programmed cell death and the failure of metabolic organs. That said, there are many other factors involved in this relationship that remain poorly understood. For example, androgen levels have been demonstrated to promote both oxidative stress and ER stress [113,114]; therefore, sex-specific differences of postnatal cell stress may exist due to altered estrogen and testosterone signaling. Furthermore, recent studies have begun to elucidate a paternal contribution to IUGR and postnatal cellular stress; paternal obesity has been demonstrated to alter placental vascular structure and postnatal liver development, likely due to the induction of ER stress in both of these organs [115]. Overall, future studies are warranted to further investigate all of these factors, and novel technologies will be required to validate the molecular findings of animal studies in human IUGR cohorts.

By better understanding the origins of cell stress and programmed cell death in IUGR offspring, postnatal therapeutic measures could be developed to reduce risk for the metabolic syndrome during adult life. Neonatal administration of exendin-4, an agonist of the glucagon-like peptide 1 receptor, has been previously shown to prevent hepatic oxidative stress in male offspring at 7–9 weeks of age, and in doing so mitigated hepatic insulin resistance [116]. The therapeutic benefits of exendin-4 have been explored in several clinical trials, and it is currently used as a treatment for type II diabetes; however, its safety and efficacy as a treatment in infants remains unknown. Similarly, the ER stress inhibitor tauroursodeoxycholic acid (TUDCA) ameliorates the incidence of type II diabetes in obese rats, so it is possible that TUDCA could be effective in treating IUGR-induced diabetes when administered in early life like exendin-4 [117]. Finally, this evolving field of research has great potential to guide clinical policy and protocols that exist during prenatal care. The cooperation of health care professionals with pregnant women is essential in preventing IUGR, and this could contribute to reduced rates of the metabolic syndrome across the adult demographic. 

## Figures and Tables

**Figure 1 ijms-22-06986-f001:**
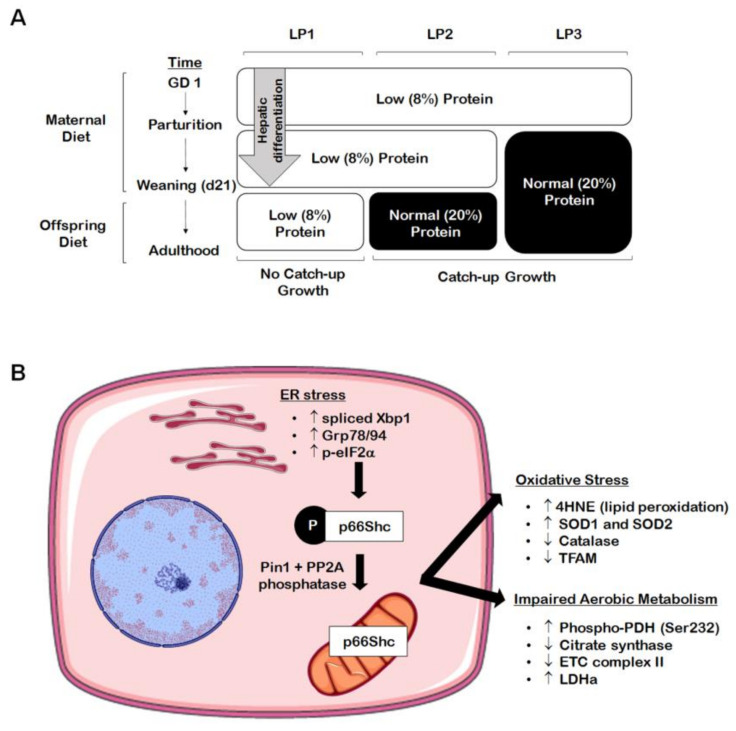
Perinatal maternal protein restriction (MPR) in combination with postnatal catch-up growth leads to endoplasmic reticulum (ER) stress, oxidative stress, and mitochondrial dysfunction in adult male rat offspring. (**A**) Percent protein composition and the timing of nutritional intervention differ across the various groups of MPR offspring ((**A**); LP1, LP2, LP3) in order to study the effects of postnatal catch-up growth on molecular outcomes in the growth-restricted liver. (**B**) Male offspring weaned onto a normal (20%) protein diet following the completion of hepatic differentiation (e.g., LP2 offspring) display increased markers of hepatic ER stress (i.e., increased spliced Xbp1, Grp78/94, and p-eIF2α) in adulthood, leading to elevation of the oxidative stress marker p66Shc. These same offspring further exhibit increased markers of oxidative stress (i.e., increased 4HNE and SOD1/2; decreased catalase and TFAM) and aberrant markers of aerobic metabolism (i.e., increased p-PDH[Ser232] and LDHa; decreased citrate synthase and complex II).

**Figure 2 ijms-22-06986-f002:**
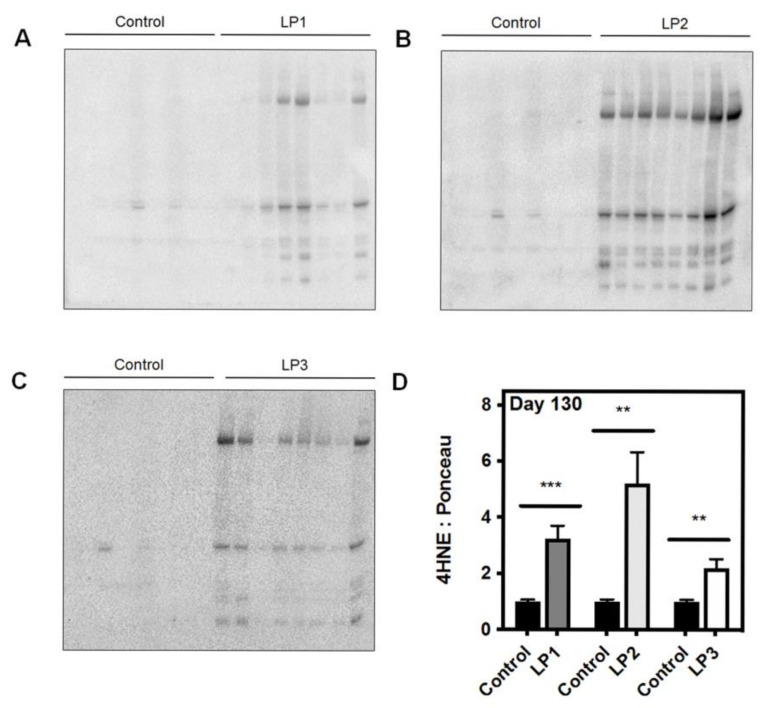
Maternal protein restriction (MPR) leads to increased hepatic lipid peroxidation in male rat offspring at four months of age. Representative western immunoblots illustrating expression of 4-hydroxynonenol (4HNE), a marker of lipid peroxidation, in (**A**) LP1, (**B**), LP2, and (**C**) LP3 offspring relative to control offspring. (**D**) 4HNE abundance for each group was compared against that of control offspring and analyzed using a two-tailed unpaired Student’s t-test. 4HNE abundance was expressed as means normalized to total protein abundance ± SEM (*n* = 7–8/group). ** Significant difference (*p* < 0.01), *** significant difference (*p* < 0.001).

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
