# Peer review of "The Role of Cellular Stress in Intrauterine Growth Restriction and Postnatal Dysmetabolism"

_ijms, 2021, doi:10.3390/ijms22136986_

Round 1
Reviewer 1 Report
This article is an up-to-date review on the role of cellular stres, particularly oxidatyive stress in intrauterine growth restriction (IUGR) and its significance for postnatal well-being. The properly selected by Authors evidence -based literature indicates the negative consequences of the growth restriction for functioning of organs, as well as of whole organism in adult life. The manuscript delivers a complete resumee of the our knowledge on the subject.
This interesting and well written paper summarizes important actual facts on IUGR, and may be published in its actual form in IJMS for the benefit of readers.
Author Response
Response to Reviewer 1 Comments
This article is an up-to-date review on the role of cellular stres, particularly oxidatyive stress in intrauterine growth restriction (IUGR) and its significance for postnatal well-being. The properly selected by Authors evidence -based literature indicates the negative consequences of the growth restriction for functioning of organs, as well as of whole organism in adult life. The manuscript delivers a complete resume of the our knowledge on the subject.
This interesting and well written paper summarizes important actual facts on IUGR, and may be published in its actual form in IJMS for the benefit of readers.
Response: We thank the reviewer for their positive comments and recognition of the importance of this topic.
Reviewer 2 Report
The presented article covers an important topic of intrauterine conditions onto the latter adverse metabolic state. The authors did a great job covering caloric and protein restriction and nicotine exposition for the developing fetus. The bibliography combines a historical overview as well as newly published manuscripts. Point 1. Could the authors please provide some additional information (if you find any) about other variants of cellular stress? Otherwise, I highly suggest reformulating the title with mentioning nicotine and food restiction. A reader might expect broader theme coverage considering the stated title. Point 2. The UPR part lacks such an obvious implication as cell cycle blockade. I believe it should be mentioned in terms of complications of dividing cells in embryonic development. Point 3. Please remove excessive abbreviations, e.g., line 55 – HPA, line 108 – AGA, line 119 – BMI, etc. They are introduced but never stated again. Such a large number of unused abbreviations is highly confusing for a reader, requiring additional working memory. Point 4. Please check the bibliography template - a couple of the references (for example, 25) do not fit it.Author Response
Response to Reviewer 2 Comments
Point 1: Could the authors please provide some additional information (if you find any) about other variants of cellular stress? Otherwise, I highly suggest reformulating the title with mentioning nicotine and food restriction. A reader might expect broader theme coverage considering the stated title.
Response 1: Thank you for this suggestion. We had initially planned to include additional forms of cell stress (i.e., the DNA damage response and the heat shock protein response), but there is limited evidence of these types during postnatal life of IUGR offspring. We agree that the manuscript does emphasize models of nutrient deprivation and drug exposure; however, much of the published literature on cellular stress in IUGR offspring is focused on models of this nature. That said, we have also commented on numerous animal models of uteroplacental insufficiency, hypoxia, preterm birth, etc., which are reflective of the human IUGR cohorts studied to date. It is for these reasons that we have opted to keep our title broader rather than focus on specific gestational insults.
Point 2: The UPR part lacks such an obvious implication as cell cycle blockade. I believe it should be mentioned in terms of complications of dividing cells in embryonic development.
Response 2: This is a great point of consideration. We have added a statement in the introductory paragraph of Section 3.2 (Endoplasmic reticulum [ER] stress and the unfolded protein response) outlining that ER stress is essential for embryonic development and the maintenance of pregnancy (lines 336–338).
Point 3: Please remove excessive abbreviations, e.g., line 55 – HPA, line 108 – AGA, line 119 – BMI, etc. They are introduced but never stated again. Such a large number of unused abbreviations is highly confusing for a reader, requiring additional working memory.
Response 3: Thank you for this suggestion, we agree that this may become confusing for a reader. We have removed any unnecessary abbreviations from the manuscript, including those mentioned in Point 3.
Point 4: Please check the bibliography template - a couple of the references (for example, 25) do not fit it.
Response 4: Thank you for bringing this up. We have gone through the list of references and corrected any individual references that do not match with the MDPI format.